# Organocatalytic Synthesis of α-Aminonitriles: A Review

**Bakhtar Ullah** [1,2,†], **Navneet Kumar Gupta** [3,*,†], **Quanli Ke** [4], **Naseeb Ullah** [2], **Xingke Cai** [2] **and Dongqing Liu** [1,*]

1 College of Mechatronics and Control Engineering, Shenzhen University, Shenzhen 518060, China
2 Institute of Advanced Study, Shenzhen University, Shenzhen 518000, China
3 Centre for Sustainable Technologies, Indian Institute of Science, Bengaluru 560012, India
4 Institute of Catalytic Reaction Engineering, College of Chemical Engineering, Zhejiang University of Technology, Hangzhou 310014, China
* Correspondence: nkgupta@iisc.ac.in (N.K.G.); liu.dongqing@szu.edu.cn (D.L.)
† These authors contributed equally to this work.

**Abstract:** α-aminonitriles, which have anticancer, antibacterial, antiviral, and antifungal properties, have played an important role in pharmacology. Furthermore, they can also be used to synthesize natural and unnatural amino acids. The main bottleneck in the commercialization of these products is their large-scale production with controlled chirality. A variety of methods have been used to synthesize α-aminonitriles. Among other reported methods for preparing α-aminonitriles, the Strecker reaction is considered appropriate. Recent developments, however, have enabled the α-cyanation of tertiary and secondary amines by functionalizing the carbon–hydrogen (C–H) bond as an attractive alternative procedure for the preparation of α-aminonitriles in the presence of an oxidant and a cyanide source. In most cases, these reactions are catalyzed by transition metal catalysts, such as Fe, Cu, Rh, V, Au, Ru, Mo, Pt, Re, and Co, or by photocatalysts. As an alternative, organocatalysts can also be used to produce aminonitriles. Although there have been numerous reviews on the preparation of α-aminonitriles, no such reviews have been published specifically on the organocatalyzed synthesis of α-aminonitriles. Organocatalysis plays a significant role in synthesizing α-aminonitriles via Strecker-type reactions and cross dehydrogenative coupling reactions (CDC). In this mini review, we discuss the organocatalyzed synthesis of these molecules. A review of new organocatalysts for the synthesis of aminonitriles is expected to provide insight into the development of new industrial catalysts.

**Keywords:** organocatalytic synthesis; α-aminonitriles; cross dehydrogenative coupling reactions; Strecker-type reactions

## 1. Introduction

α-aminonitriles are widely used in various fields of pharmacology (Figure 1). Firstly, they are the starting materials for the synthesis of amides, 1,2-diamines, α-amino acids, α-amino alcohols, and α-amino carbonyls, as well as imidazoles and thiadiazoles, all of which are basic and important units in the field of synthetic chemistry [1–6]. Secondly, α-aminonitriles containing compounds play significant roles in various therapeutic activities, such as anticancer, antiviral, and antibacterial [1,2]. For the treatment of diabetes, some aminonitriles have been recognized as reversible inhibitors of dipeptidyl peptidase (DPP4) [7]. Nitriles of *N*-aryl-tetrahydroisoquinolines (THIQs) have been employed as pesticides against animal mites, such as *Psoroptes cuniculi* [8]. Other aminonitriles containing compounds, such as phthalascidin and saframycin A, show effective anti-tumor activity [1,2]. Moreover, (S)-clopidogrel and antiplatelet drug, can be prepared from 2-(2-Chlorophenyl)-2-(6,7-dihydrothieno [3,2-c]pyridine-5(4H)-yl)acetonitrile [9]. Therefore, the synthesis of α-aminonitriles has received much attention in the pharmaceutical field.

**Figure 1.** General structural types of α-aminonitriles. Representative biologically active α-amino nitrile-containing natural products and synthetic drugs. The general structure of these biologically active α-amino nitriles is represented in the highlighted box, where $R^1$, $R^2$, $R^3$, and $R^4$ = alkyl, aryl [2]. Reprinted with permission from Ref. [2]. Copyright 2018, Elsevier Ltd.

The synthesis of α-aminonitriles was first reported in 1850 by mixing an aqueous solution of ammonia, acetaldehyde, and hydrogen cyanide (HCN), giving the corresponding product of α-aminonitrile [1,2,10–13]. This method is named as Strecker reaction and has been recognized as one of the oldest multi-component condensation reactions [1,2]. Until now, the Strecker reaction is still the most applied and effective method for the preparation of α-amino acids from the α-aminonitriles [1,2]. In order to increase the production yield of α-aminonitriles efficiently, several synthetic methods have been developed in this regard. A wide range of cyanation agents, such as KCN, HCN, TMSCN, $CH_3COCN$, $Bu_3SnCN$, and ethyl cyanoformate, have been studied to optimize the results. On the other hand, different types of catalysts, such as metal-catalysts, bio-catalysts, and organocatalysts, have been used for α-aminonitriles preparation [2]. Although metal catalysis is dominant in the synthesis of α-aminonitriles [1,2], both in the Strecker-type reactions and in CDC reactions via C–H bond activation of tertiary or secondary amines, but, metal catalysts are environmentally unfriendly, expensive, and unsafe for use in pharmaceutical industries [14,15]. In the case of bio-catalysis problem arises, such as enzyme growth, instability, and separation [16,17]. Therefore, in recent decades, organocatalysis has received considerable attention as an environmentally benign method in synthetic chemistry [14,15,18]. Compared to the toxic metal-based catalysts [15], the organocatalysts are less harmful and more suitable for use in pharmaceutical industries [14,15,18]. In addition, the organocatalysts have other advantages, such as easy availability, good efficiency, low cost, good selectivity, and less sensitivity to air and moisture [14,15,18]. Therefore, a review of the current organocatalytic methods applied for the synthesis of α-aminonitriles is very necessary and should be helpful to the development of this important field in pharmaceutical sciences.

Organocatalysis is described as the acceleration of chemical reactions by the sub-stoichiometric amount of small molecules having no metal active sites [14,18,19]. During the past two decades, organocatalysis has evolved into a practical synthetic paradigm and has received much scientific research interest in chemical synthesis due to its properties,

such as operational simplicity, ready availability, excellent moisture resistance, etc. [14,15]. The reactions catalyzed by organocatalysts are becoming important routes in the synthesis of complex molecular skeletons [14,18]. In the meantime, organocatalysis has also been taken as an efficient synthetic process for providing alternative and environmentally benign routes for chemical synthesis, avoiding the use of toxic metal catalysts. In parallel with enzymatic and metal catalysis, organocatalysis has now become a dominant catalytic approach in synthetic chemistry and has opened a new avenue for the construction of new carbon–carbon/carbon–heteroatom/heteroatom–heteroatom (C–C/C–X/X–X) bonds. Most recently, the inert C–H bond functionalization is one of the prominent developments in organocatalyzed reactions as it is a potential synthetic approach for (C–C/C–X/X–X) bond synthesis without the pre-functionalized groups as needed in the conventional strategies.

Organocatalysts have been used successfully to develop versatile synthetic methods for α-aminonitriles, such as the anodic oxidative cyanation of tertiary amines, the Strecker reaction, and the oxidative cyanation of tertiary or secondary amines. Two practical and effective methods have been chosen to synthesize α-aminonitriles: Strecker-type reactions and CDC reactions via oxidative C–H bond activation of tertiary or secondary amines. This mini-review will focus on the progress of synthesizing α-aminonitriles using these two strategies, as well as the advantages and disadvantages of each method. Finally, an outlook for future opportunities and insight into current challenges regarding α-aminonitrile synthesis will be presented.

## 2. Organocatalyzed Synthesis of α-Aminonitriles via Strecker-Type Reactions

The Strecker reaction is a classical one-pot multi-component reaction, which was developed by Strecker in 1850 by simple mixing of acetaldehyde, hydrogen cyanide, and ammonia to produce α-aminonitriles [1,2]. The acid hydrolysis of amino nitrile produced from this method was used to synthesize the first amino acid, alanine [1,2,20]. Therefore, the Strecker reaction attracted significant attention from chemists after its discovery, and it is still very popular today [1,2]. Encouraged by the increasing demand for amino acids in a wide range of fields, including chemistry, life sciences, and different industrial applications, asymmetric Strecker reaction has been one of the popular topics in recent years in organic chemistry.

Generally, the chiral auxiliaries are required in stoichiometric amounts to proceed with the asymmetric Strecker reactions and are needed to remove from the resulting products. There are two methods applied for asymmetric Strecker reactions to achieve enantioenriched α-amino nitriles. The first method is based upon the cyanide addition to chiral imines, where the chiral imine moieties arise from ketones/aldehydes or/and amines. By using optically pure chiral amines as the chiral auxiliary for highly asymmetric Strecker reaction, it was able to produce a range of enantiomerically pure α-amino acids. The second method is based on the enantioselective catalytic cyanation of achiral imines.

Later, a remarkable development in this method was reported by K. Harada in 1963 [1,21]. K. Harada applied enantioenriched (S)-R-phenylethylamine in the typical Strecker reaction instead of ammonia [1,21]. Although the Strecker reaction has generally substrate scope limitation to aldehydes, it has the advantage of preparing the α-aminonitriles by providing a direct, robust, and economically practical route to synthesize a wide range of naturally and non-naturally α-amino acids [1]. In this section, we will divide the synthesis of α-aminonitriles by Strecker reaction into two parts: (1) synthesis of chiral α-aminonitriles and (2) synthesis of racemic α-aminonitriles.

### 2.1. Synthesis of Chiral α-Aminonitriles

A molecule can be considered to be chiral if it has an isomer that has a mirror configuration of itself. In chiral molecule, the importance of isomerism is not limited to its ability to rotate plane-polarized light in opposite directions, but it goes far beyond this. A large number of molecules produced by organisms possesses a particular handedness. A particular molecule often has a particular effect on an organism based on how it binds to a receptor

molecule within the organism. Left-handed receptors require a particular enantiomer for a proper fit, just like a glove designed for a left hand. For this reason, when designing pharmaceuticals, we must consider which enantiomer will fit the intended receptor.

In this part, we will discuss the recent advances in the organocatalyzed Strecker synthesis of chiral α-aminonitriles. An organocatalytic Strecker reaction of ketones using Brønsted acids was developed by Guang-Wu Zhang and coworkers in 2010. Several α-aminonitriles were obtained in yields of 79–99%. Moreover, they extended their work for enantioselective reactivity of the three-component Strecker reaction and achieved a yield of up to 40% [22]. In 2012, S. Saravanan et al. reported a recyclable chiral amide-based organocatalyst for the asymmetric Strecker reaction (Scheme 1). The desired α-aminonitriles products were obtained in high yields with excellent *ee* up to 99%. They also used this catalytic system for the synthesis of high *ee* of (R)-3-phenylpropane-1,2-diamine and (R)-phenylalanine. The catalyst was recycled and reused for three runs without affecting yields and *ee* of products, which makes the catalyst more convenient for practical applications [23]. The proposed mechanism was based on experimental investigations in which it was concluded that the imine nitrogen of the substrate and the sulfonamide proton of the catalyst interacts to make an intermediate A in the catalytic cycle (Scheme 2). Afterward, the addition of a cyanide source (ethyl cyanoformate), led to an intermediate B, which provides the resultant products by addition of isopropyl alcohol (IPA), as shown in Scheme 2 [23].

In 2012, Arghya Sadhukhan et al. approached an asymmetric Strecker reaction with aldehydes and secondary amines for the construction of α-aminonitriles using hydroquinine as a chiral catalyst in the presence of sodium fluoride. This catalytic system afforded several essential α-aminonitriles in excellent yields with high enantioselectivities, given in Scheme 3 [24].

Subsequently, *S*-clopidogrel (an antiplatelet reagent) was efficiently prepared from the analogous α-aminonitrile (2-(2-Chlorophenyl)-2-(6,7-dihydrothieno [3,2-c]pyridin-5(4H)-yl)acetonitrile), as illustrated in Scheme 4 [24]. *S*-clopidogrel is an antiplatelet drug used to decrease the danger of heart disease and stroke in those at considerable risk [9].

**Scheme 1.** Chiral amide-based organocatalyst **5** catalyzed asymmetric Strecker reaction. R is (substituted) aryls or alkyls [23]. Reprinted with permission from Ref. [23]. Copyright 2012, American Chemical Society.

○ = for the sake of clarity only half of the unit is shown

**Scheme 2.** The probable mechanism for enantioselective Strecker reaction catalyzed by chiral amide-based organocatalyst **5** [23]. Reprinted with permission from Ref. [23]. Copyright 2012, American Chemical Society.

**Scheme 3.** Hydroquinine catalyzed enantioselective Strecker reaction. R is (substituted) aryls or alkyls [24]. Reprinted with permission from Ref. [24]. Copyright 2012, American Chemical Society.

**Scheme 4.** Synthesis of (S)-clopidogrel from the relevant α-aminonitrile [24]. Reprinted with permission from Ref. [24]. Copyright 2012, American Chemical Society.

In 2012, Hailong Yan and co-workers developed a 3,3′-diiodine-substituted BINOL (1,1′-bi-2,2′-naphthol)-based organocatalytic protocol for asymmetric Strecker reaction that uses a readily available chiral version of oligoethylene glycol as a catalyst and KCN to make a chiral cyanide anion as demonstrated in Figure 2. The metal ($K^+$) present at the active organocatalyst site is provided by the KCN reagent. This method provided excellent yields and efficiency for the synthesis of asymmetric α-aminonitriles Strecker products from α-amido sulphones. Using this efficient catalytic protocol, enantiomerically pure non-natural a-amino acids were produced on a gram scale from α-amido sulphones [25].

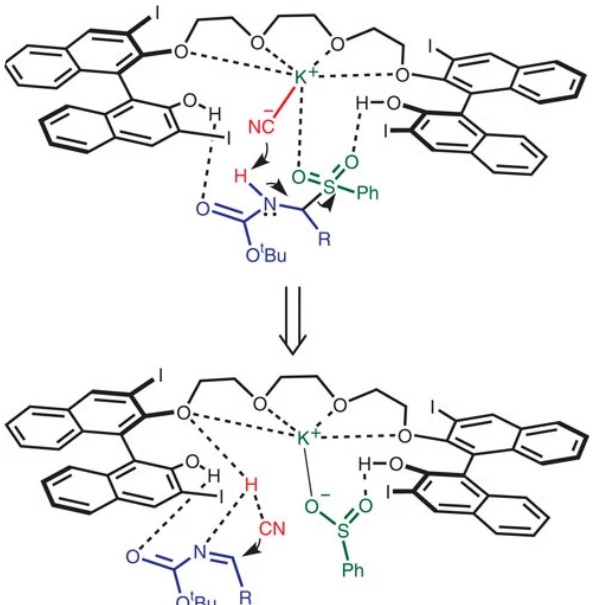

**Figure 2.** Chiral bis-hydroxy polyether as a chiral anion generator [25]. Reprinted with permission from Ref. [25]. Copyright 2012, Springer Nature Limited.

A plausible mechanism was showing that the anion (cyanide) group of the catalyst attracts proton from α-amido sulphone converting it to imine. Subsequently, the cyanide group from the in situ generated HCN again attack the imine carbon converting it to the corresponding α-aminonitrile, see Scheme 5 [25].

**Scheme 5.** The mechanism proposed for the asymmetric Strecker reaction [25]. Reprinted with permission from Ref. [25]. Copyright 2012, Springer Nature Limited.

In 2013, S. Saravanan et al. introduced a small molecule as a chiral organocatalyst for an enantioselective Strecker reaction. Various substrates derivatives of *N*-benzhydryl- and *N*-tosyl-substituted imines were tolerated well with this chiral catalyst (based on sulfinamide motif) in the presence of *i*-PrOH as an additive using ethylcyanoformate as a cyanide source affording the analogous chiral α-aminonitriles with high yields and excellent *ee* up to 99% at 0 °C in 24–30 h. Additionally, this homogenous catalyst was recovered and reused for three turns without any loss in its efficiency and enantioselectivity [26].

The proposed mechanism was supported by experimental results, showing that the catalyst works as Jacobsen's thiourea [27] system, i.e., the –NH groups of the catalyst activate the imine and the cyanide source ethylcyanoformate simultaneously. In turn, the cyanide group attacks the imine carbon to afford the corresponding α-aminonitriles [26].

In 2013, Arghya Sadhukhan and co-workers developed an oxazoline-based organocatalyst for asymmetric Strecker reaction of various aromatic and aliphatic *N*-benzhydrylimines at −20 °C affording the relevant α-aminonitriles in excellent yields and *ee* up to 96% and 98%, respectively. They successfully extended this organocatalytic protocol to obtain a high yield with a high *ee* of pharmacologically significant drug molecule levamisole [28].

To justify the enantioselective construction of the product, DFT calculations have been performed with the organocatalyst in these reactions. The catalyst has a unique design having different functional groups [28].

In 2014, Hai-Xiao He and Da-Ming Du developed a cinchona-based squaramide organocatalyst for chiral Strecker reaction of imines having a thiazole moiety acquiring the resultant products in good to excellent yields with excellent *ee* up to 98% [29]. In the mechanism, the heteroaromatic imine is activated by squaramide organocatalyst by a hydrogen bond between the two NH groups of the squaramide catalyst and the two N atoms of the heteroaromatic imine. The basic quinuclidine moiety of the squaramide catalyst activates the in situ generated HCN obtained from TMSCN. A nucleophilic cyanide group attacks the C=N double bond of imine, yielding the analogous a-aminonitriles [(S) configuration] with excellent enantioselectivity [29].

In 2017, Amamudin Ansari et al. developed a recyclable quinine thiourea organocatalyst anchored on the Santa Barbara Amorphous–15 (SBA–15) mesoporous material for enantioselective Strecker reaction of isatin *N*-protected ketimines. By using this heterogeneous catalyst, $\alpha$-aminonitriles were obtained in good yields and high up to 93% at $-10\ ^{\circ}$C in 85 h [30]. The recyclability of the solid catalysts was tested by performing cycle of reactions after separation from first run [30]. The recovered catalyst showed excellent activity up to five cycles with retention of *ee* around 91%. However, the yield was decreased to 85–90% due to entrapment of substrate in the catalyst. The organocatalyst was quite stable on surface which was proved by the ICP analysis of reaction solution.

In 2017, Suruchi Mahajan et al. documented an enantioselective Strecker reaction catalyzed by pseudo-enantiomeric squaramide catalysts for chiral $\alpha$-aminonitriles preparation giving good yields with high enantioselectivity (Scheme 6). It was proposed that the H-bonding of squaramide moiety is involved in the reaction mechanism [31].

**Scheme 6.** Organocatalytic enantioselective Strecker reaction with ketimines catalyzed squaramide catalysts [31]. Reprinted with permission from Ref. [31]. Copyright 2017, Royal Society of Chemistry.

Recently, Kadota et al. reported Strecker reaction of isatin-derived N-unsubstituted ketimines to N-unprotected $\alpha$-aminonitriles with tetrasubstituted carbon stereocenter up to 99% *ee* [32]. This reaction does not require the protection/deprotection steps. This protocol has been useful for the synthesis of unnatural amino acids via one-pot stereoselective routes towards biologically active compounds.

### 2.2. Synthesis of Racemic α-Aminonitriles

There are a few reports that catalyst-free protocols promote the Strecker reactions for the synthesis of α-aminonitriles. However, these catalyst-free protocols have various limitations in one or another aspect, i.e., substrate scope limitation, lower yields, longer reaction times, etc. [33–36].

In 2011, M.G. Dekamin et al. introduced nano-ordered mesoporous borosilicate (B-MCM-41) for three-component Strecker reaction for α-aminonitriles production from a variety of aldehydes and ketones with amines and TMSCN. The relevant products were afforded high to excellent yields at mild reaction conditions [37].

In 2012, Devdutt Chaturvedi et al. described a Strecker reaction for efficient synthesis of α-aminonitriles using Mitsunobu's reagent, i.e., an equimolar mixture of triphenylphosphine and diethyl azodicarboxylate, as a catalyst in a solvent-free system. Various α-aminonitriles were prepared in high yields (80–99%) [38].

In 2013, Dariush Saberi and fellows developed a dehydroascorbic acid capped magnetite (DHAA-$Fe_3O_4$) catalyst for Strecker reaction to synthesize α-aminonitriles and α-aminophosphonates. Various derivatives of the relevant products were synthesized in good yields at ambient reaction conditions, see Scheme 7. The catalyst was reused six times in a model reaction of benzaldehyde, aniline, and TMSCN without any significant loss in the catalyst reactivity [39].

**Scheme 7.** DHAA-$Fe_3O_4$ catalyzed Strecker reaction for α-aminonitriles formation. $R^1$ is various (substituted) aryls or alkyls and $R^2$ is H or Me [39]. Reprinted with permission from Ref. [39]. Copyright 2013, Elsevier Ltd.

The three-component Strecker reaction mechanism was proposed in multiple steps (Scheme 8). The DHAA-$Fe_3O_4$ nanoparticles promote the in situ construction of the imine intermediate via the formation of hydrogen bonding between the carbonyl group oxygen and the DHAA hydroxyl groups. Then, the amine attacks the activated carbonyl carbon generating imine with water production. Subsequently, the cyanide group from TMSCN attacks the imine carbon to provide the desired product, as illustrated in Scheme 8 [39].

In 2013, Aayesha Nasreen reported *L*-proline as an organocatalyst for a one-pot three-component Strecker reaction to synthesize a series of α-aminonitriles from the condensation of aldehydes, amines, and TMSCN providing good to excellent yields (72–95%) at ambient temperature in acetonitrile as a reaction medium [40].

In 2014, Hossein Ghafuri et al. presented aqueous formic acid as a catalyst in the Strecker reaction to afford α-aminonitriles and imines. The corresponding products were obtained in high yields at room temperature [41]. In the proposed mechanism, it was assumed that the carbonyl group of aldehydes was activated via hydrogen bonding by the acidic hydrogen of formic acid, which, in turn, was susceptible to the nucleophilic attack of amines to give imines. Then, the imines were attacked by the CN group to produce the desired products [41].

**Scheme 8.** The proposed mechanism for the three-component Strecker reaction catalyzed by DHAA-Fe$_3$O$_4$. R$^1$ is various (substituted) aryls or alkyls and R$^2$ is H or Me [39]. Reprinted with permission from Ref. [39]. Copyright 2013, Elsevier Ltd.

In 2014, Baskar Nammalwar et al. synthesized $\alpha$-aminonitriles by Strecker reaction in three ways; i.e., (a) by employing 3 mol % of NH$_4$Cl as a catalyst in ethanol with TMSCN (1.0 equivalent) followed by conventional heating (b) with a little increase in TMSCN (1.2 equivalent) in ethanol followed by conventional heating and (c) microwave heating with a similar excess of TMSCN in the absence of any solvent medium. All methods afforded the resultant $\alpha$-aminonitriles in good to high yields; however, the microwave heating method was superior to other methods giving a number of the resultant $\alpha$-aminonitriles in high to excellent yields just in 30–240 s [42].

In 2017, Krishna S. Indalkar and co-workers labeled a three-component Strecker reaction for $\alpha$-aminonitriles preparation from aldehydes/ketones, amines, and TMSCN under solvent-free reaction conditions at room temperature. This reaction catalyzed by sulfated polyborate gave excellent yields up to 99% of respective $\alpha$-aminonitriles having different functional groups, see Scheme 9. There was no significant loss in catalytic efficiency of the catalyst after reusing four times in a model reaction of benzaldehyde with aniline in the presence of TMSCN as a cyanation reagent [43].

**Scheme 9.** Strecker reaction catalyzed by sulfated polyborate for $\alpha$-aminonitriles construction. R$^1$ = various (substituted) aryls or alkyls, R$^2$ = H or Me, R$^3$ = (substituted) aryls or alkyls, R$^4$ = H or alkyls [43]. Reprinted with permission from Ref. [43]. Copyright 2017, Elsevier Ltd.

In 2017, Saeed Baghery and co-workers reported four magnetic nanoparticle (MNP) catalysts with urea or urethane moieties for Strecker synthesis of α-aminonitriles, given in Scheme 10. The magnetic nanoparticles coated with silica were simply activated by the addition of 3-(triethoxysilyl)propylisocyanate (TESPIC), amine, or amino alcohol. The TESPIC having dual labile functional groups was employed as an applicable precursor for the construction of urethane-based catalysts. These recyclable functionalized catalysts were employed in the synthesis of α-aminonitriles giving high to excellent yields at 50 °C under solvent-free conditions. To check the recyclability of the catalyst, the reaction was monitored by TLC thoroughly. When the reaction was completed the catalyst was separated by an external magnet, after washing with ethanol, reused seven times without any change in its efficiency [44].

**Scheme 10.** Strecker reaction catalyzed by MNP catalysts with urea or urethane moieties. R′ = (substituted) aryls or alkyls, R″ = (substituted) aryls or alkyls [44]. Reprinted with permission from Ref. [44]. Copyright 2017, John Wiley & Sons, Inc.

In the proposed mechanism, the carbonyl group is activated by hydrogen-bonding of amide groups of the catalysts, making it susceptible to the nucleophilic attack of the amine to produce imine. In turn, imine nitrogen is further activated by the amide group of the catalyst via hydrogen bonding. Then, the imine carbon is attacked by the cyanide group to afford the corresponding adduct of α-aminonitrile [44].

Recently Ghogare et al. proposed succinic acid as a noble and efficient organocatalyst for the synthesis of α-aminonitrile under solvent free conditions [45]. Succinic acid as a catalyst resulted high conversion and isolated yields more than 90%. Acidic proton of succinic acid was proposed as main active site for the activation of carbonyl carbon of aldehyde for the efficient reaction with amines to form imine. In last step nucleophilic addition between imines and trimethylsysil cyanide led to the α-aminonitrile.

### 3. Organocatalyzed Synthesis of α-Aminonitriles via Oxidative C–H Bond Functionalization

In this section, we will describe organocatalyzed oxidative synthesis of α-aminonitriles. As we have seen earlier that Strecker-type reactions are frequently used for the preparation of α-aminonitriles. However, certain organocatalyzed procedures exist with respect to the oxidative synthesis of α-aminonitriles. Despite their inert nature, C($sp^3$)–H bonds have a lot of potential for selective functionalization in organic synthesis and catalysis. Although catalytic C($sp^3$)–H functionalization is useful for converting hydrocarbons lacking directing groups into chiral molecules, it is a formidable task. As the synthesis of α-aminonitriles via C–H bond functionalization is introduced recently, and it will take time to stand in parallel with the Strecker-type reactions.

In 2014, Hirofumi Ueda and co-workers developed a catalytic protocol of acetic acid with molecular oxygen as an oxidant for the oxidative activation of the benzylic C–H bonds in *N*-aryltetrahydroisoquinoline (THIQs) and tetrahydro-β-carboline derivatives

to construct various C–C bonds including α-aminonitriles. A variety of corresponding products were afforded by this catalytic protocol in moderate to high yields at 70 °C [46].

In the proposed mechanism the reaction goes forward via a benzylic radical species Scheme 11B generated through the auto-oxidation of Scheme 11A. The molecular oxygen reacts with species Scheme 11B creating species Scheme 11C which abstracts the benzylic proton from another molecule of substrate Scheme 11A generating species Scheme 11D. In the next step, hydrogen peroxide elimination from Scheme 11D would give the iminium ion Scheme 11E, which, in turn, would be able to make a new C–C bond using different nucleophiles including cyanide anion, giving the analogous product Scheme 11F, as shown in Scheme 11. With 2,6-di-*tert*-butyl-4-methylphenol (radical inhibitor), a control experiment suggests the involvement of the radical species Scheme 11B, as the reaction conversion was much inferior with radical inhibitor than the reaction without radical inhibitor [46].

**Scheme 11.** The plausible mechanism of acetic acid oxidative catalyzed the synthesis of α-aminonitriles [46]. Reprinted with permission from Ref. [46]. Copyright 2014, American Chemical Society.

Our 2019 investigation reported the use of thiourea as a mild and effective organocatalyst to oxidatively a-cyanate N-aryltetrahydroisoquinolines (THIQ) with cyanide source TMSCN, yielding a-aminonitrile analogs in good to excellent yields (Scheme 12). The results of this study show that thiourea acts as a radical initiator by abstracting hydroxyl radicals (*OH) directly from tert-butyl hydroperoxide (TBHP) rather than through noncovalent hydrogen bondings (H-bonds) [47].

**Scheme 12.** The substrate scope of thiourea catalyzed oxidative α-cyanation of tertiary amines. $R^1$ = aryl, heteroaryls; $R^2$ = H, OMe; $R^3$ = H, OMe [47]. Reprinted with permission from Ref. [47]. Copyright 2019, Elsevier Ltd.

Thiourea catalyzed oxidative cyanation of THIQs is not fully understood. According to our results, tetramethylthiourea can also enhance THIQ oxidative cyanation with TMSCN in the presence of TBHP, suggesting that H-bonding might not play a crucial role. Considering

the reactivity of tetramethylthiourea, Scheme 13 proposes a plausible mechanism. The H-bonding interactions between thiourea derivatives and TBHP aid in the construction of an active iminium intermediate [48]. However, thiourea likely acts as a radical initiator by accommodating hydroxyl radicals (*OH) from TBHP to form tert-butoxyl radicals (t-BuO). As a result of the reaction involving the tert-butoxyl radical, a SET from substrate 1 forms a radical ion pair consisting of the N-aryltetrahydroisoquinoline radical cation 4 and the tert-butoxyl anion. As an intermediate, radical cation 4 is transferred to thiourea-hydroxyl radical, producing iminium ion 5, while thiourea is reconstituted as water ($H_2O$). The iminium ion 5 go through a nucleophilic attack of HCN (in situ generated from in methanol) to afford the desired product 2, as illustrated in Scheme 13 [47].

**Scheme 13.** The proposed mechanism of thiourea catalyzed oxidative cyanation of THIQs [47]. Reprinted with permission from Ref. [47]. Copyright 2019, Elsevier Ltd.

An oxidative cyanation of tertiary amines was recently reported by Peng-Yu Liu and coworkers using azobis(isobutyronitrile) (AIBN) as cyanide source with pivalic acid and sodium acetate as additives at ambient reaction conditions. By this method, it was possible to obtain moderate to high yields of the corresponding $\alpha$-aminonitriles [49].

Hang Shen et al. developed an oxidative $\alpha$-cyanation method for preparing a-aminonitriles from tertiary amines catalyzed by in situ generated intermediates of PIFA (or DIB) and TMSCN in 2015. This catalytic protocol tolerates a wide range of substrates to produce the relevant products in low to high yields [50].

Recently base mediated C-H bond cyanation for the synthesis of $\alpha$-aminonitriles from sulfonamides is reported by Shi and coworkers [51]. They developed an effective and metal-free synthetic protocol containing N-fluorotosylamides, TMSCN, $Et_3N$ in toluene is capable of $\alpha$-C–H bonds dehydrocyanation of amides at 40 °C to achieve excellent yields up to 90%. Broad applicability of this reaction system was proved by taking various complex molecules to achieve high isolated yields [51].

**4. Summary and Outlook**

$\alpha$-aminonitriles are valuable synthons and have played a vital role in various fields, such as pharmaceutical science and agro-industries. As we have seen in the literature, most organocatalysts employed for synthesizing $\alpha$-aminonitriles have some limitations, i.e., low yields, substrate scope limitation, extended reaction time, non-recyclability, and, more often, the production of racemic products. We need chiral products to use in practical fields instead of racemic mixtures. To achieve this, developing chiral organocatalysts is crucial for both Strecker-type reactions and oxidative cross-coupling reactions for producing asymmetric $\alpha$-aminonitriles.

Organocatalyzed Strecker-type reactions and oxidative cross-coupling reactions are attractive ways of synthesizing a diverse range of α-aminonitriles. The Strecker-type reactions are dominated by oxidative cross-coupling reactions for synthesizing α-aminonitriles, even more specifically for asymmetric versions, as Strecker-type reactions have a long history in their design development. The organocatalyzed oxidative cross-coupling reactions have been newly introduced in the last few years. Currently, they have been used only for preparing the racemic version of α-aminonitriles. There is always a risk of anticipating the future of any domain. However, numerous features of chiral organocatalysis will certainly attract researchers interested in developing asymmetric organocatalysts. With time, we will continue towards the design and discovery of chiral organocatalysts having new reactivities, better proficiency, excellent *ee*, recyclability, and higher turnover numbers for the oxidative synthesis of asymmetric α-aminonitriles. To get rid of these problems in Strecker-type and oxidative cross-coupling reactions, we need to design and prepare chiral organocatalysts with accumulative properties of practical catalysis.

Although the contribution to this exciting research field, there are many pathways, for example, developing asymmetric catalytic approaches, trying new one-pot three-component strategies, discovering novel organocatalysts, and using different cyanide sources. The main target is synthesizing chiral α-aminonitriles more practically, efficiently, economically, conveniently, and eco-friendly, with greater yields and higher *ee* at ambient reaction conditions, both by Strecker-type reactions and oxidative cross-coupling reactions. In the future, it may be feasible to graft chiral organic ligands onto surfaces of 2D materials, such as graphene, h-BN, black phosphorus, etc. That would offer the advantages of recyclability, retention chirality, non-toxicity, ease of handling, and ease of separation of the reaction mixture. All these properties would turn it into metal-free catalysts suitable for practical applications. We hope that cheaper, efficient, and readily available asymmetric organocatalysts with novel ideologies will be available in the near future, giving greater yields and higher *ee* in shorter reaction times at ambient reaction conditions. We hope this insight will be helpful in this exciting and stimulating area.

**Author Contributions:** Conceptualization, B.U., N.K.G., X.C. and D.L.; funding acquisition, D.L. and N.K.G.; writing—original draft preparation, B.U. and N.K.G.; writing—review and editing, N.K.G., Q.K., N.U., X.C. and D.L. All authors have read and agreed to the published version of the manuscript.

**Funding:** This work was supported by Natural Science Foundation of China (No. 22105129), Guangdong Basic and Applied Basic Research Foundation (No. 2022A1515011048), and NTUT-SZU Joint Research Program (No. 2022005). N.K.G. gratefully acknowledges financial support from the Indian Institute of Science.

**Data Availability Statement:** Data sharing is not applicable to this article. No new data were created or analyzed in this study.

**Conflicts of Interest:** The authors declare no conflict of interest.

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
