# Peer review of "Organocatalytic Synthesis of α-Aminonitriles: A Review"

_catalysts, doi:10.3390/catal12101149_

Round 1

Reviewer 1 Report

Although the manuscript has been written well yet there are some which need to be removed:

1 - The numbering of the compounds is not uniform. Somewhere there are numbers present and somewhere there are no numbers in the structures. There need to have some consistency in the numbering of the compounds throughout the manuscript.

2   - Each synthetic scheme should have the reaction yields and stereoselectivities. If a general reaction scheme has been used such as scheme 1/scheme 7scheme 9/ scheme 10, overall reaction yields and stereoselectivities should be mentioned. In such cases, R should also be specified.

3   - In scheme 1, all the reaction conditions along with compound numbers are bold. Only compound numbers should be bold only.

4    -  In scheme 4, yield of 6 should be mentioned.

5.      In scheme 11, the presentation of the mechanism should be improved.

6.      All the equations in scheme 13 should be merged together to make one.

7. Conclusion should be concise

Author Response

Although the manuscript has been written well yet there are some which need to be removed:

1 - The numbering of the compounds is not uniform. Somewhere there are numbers present and somewhere there are no numbers in the structures. There need to have some consistency in the numbering of the compounds throughout the manuscript.

Answer: Thanks for the positive comment. We appreciated the useful suggestions regarding the inconsistency in the numbering of the compounds. In the revised manuscript we corrected all the figures to make them more consistent with numbering.

2   - Each synthetic scheme should have the reaction yields and stereoselectivities. If a general reaction scheme has been used such as scheme 1/scheme 7scheme 9/ scheme 10, overall reaction yields and stereoselectivities should be mentioned. In such cases, R should also be specified.

Answer: As suggested by the reviewer the reaction yields and stereoselectivities are presented in the revised manuscript. Also, the schemes are modified for more clarity, and reaction yields and stereoselectivities are added. R is given in the corresponding figure if necessary.    

3   - In scheme 1, all the reaction conditions along with compound numbers are bold. Only compound numbers should be bold only.

Answer: Scheme 1 was copied from the cited literature. However, based on the reviewer’s comment we modified Scheme 1 and removed the numbering which was not important to the review. Additionally, we added the reaction yield and stereoselectivity.

4    -  In scheme 4, yield of should be mentioned.

Answer: Scheme 4 is made clearer by removing the numbering of molecules. The yield of the first step is presented; however, the total reaction yields, or reaction of the second step is not clearly maintained. Therefore, we could not include the yield of the final product.

  1. In scheme 11, the presentation of the mechanism should be improved.

Answer: Thank you for the comment. We rechecked all graphics for readability and design and adapted where suitable.

  1. All the equations in scheme 13 should be merged together to make one.

Answer: Scheme 13 is further modified based on the clarity of the mechanism.

  1. Conclusion should be concise

Answer: Thank you for your comment. This was considered for all sections based on the constructive comments given by reviewers.

The comments from three reviewers were most helpful and gave us a better perspective of our work. We feel that the present manuscript is greatly improved by the revisions made. Now we consider that this paper will be of interest to the many researchers reached by Catalysts.

Reviewer 2 Report

This is a valuable review article with authors summarizing the key findings for the synthesis of -aminonitriles. Indeed, this topic is of interest in the light of the last Chemistry Nobel and overall authors have comprehensively summarized key findings consistently, both from technical and non-technical standpoints. 

This article can be accepted for publication in Catalysts with minor revisions listed below. 

1.     Authors should make list of references more comprehensive, and current. There seem to be only 4 citations from 2020 and after so kindly include some of new discoveries even they aren’t specifically covered in this work. A reasonable number of citations for this length of review article should be at least 50. 

2.     Authors must also define acronyms as extensively as appropriate, for instance isopropanol must be defined as IPA before being used consequently and consistently. 

3.     Please clarify the footnote of Figure 1 and specify that general structural types are shown in the box. 

4.     Authors could also show mechanistic steps more clearly, this applies to almost all the schemes. In scheme 2 for instance, ethyl cyanoformate could be shown once and the nucleophilic attack of the cyano moiety on the imine could be shown clearly by arrow moving toward N center of the –C=N–. If possible, kindly remediate such issues throughout the manuscript.  

5.     On a side note, and not entirely pertinent to this article, there are already work where chiral ligands have been ionically and covalently tethered to heterogeneous surfaces and authors have raised a good point highlighting this in the conclusion section that more work can be done in this area.

Author Response

This is a valuable review article with authors summarizing the key findings for the synthesis of -aminonitriles. Indeed, this topic is of interest in the light of the last Chemistry Nobel and overall authors have comprehensively summarized key findings consistently, both from technical and non-technical standpoints.

Answer: Thanks for the positive comments. -aminonitriles are an essential class of chemicals with application in many fields, especially in pharmaceutical industries. We strongly believe this review will be a key summary for researchers in this area.

This article can be accepted for publication in Catalysts with minor revisions listed below. 

  1. Authors should make list of references more comprehensive, and current. There seem to be only 4 citations from 2020 and after so kindly include some of new discoveries even they aren’t specifically covered in this work. A reasonable number of citations for this length of review article should be at least 50. 

Answer: Thanks for the comment. After 2020 the publication is quite limited in this field therefore citations are few afterward. As suggested, we have included new discoveries related to the topic of this review to make citations over 50. Additionally, we added a few recent literatures in sections 2.2 and 3.

  1. Authors must also define acronyms as extensively as appropriate, for instance isopropanol must be defined as IPA before being used consequently and consistently.

Answer: Thanks for the suggestion. We worked on the acronyms throughout the manuscript and corrected them for clarity.

  1. Please clarify the footnote of Figure 1 and specify that general structural types are shown in the box.

Answer: We corrected the footnote of Figure 1 of the revised manuscript.

Page 2, lines 46-48

Figure 1. General structural types of -aminonitriles. Representative biologically active -amino nitrile-containing natural products and synthetic drugs. The general structure of these biologically active -amino nitriles is represented in the highlighted box.2.

  1. Authors could also show mechanistic steps more clearly, this applies to almost all the schemes. In scheme 2 for instance, ethyl cyanoformate could be shown once and the nucleophilic attack of the cyano moiety on the imine could be shown clearly by arrow moving toward N center of the –C=N–. If possible, kindly remediate such issues throughout the manuscript.

Answer: As suggested by the reviewer we rechecked all graphics for readability and design and adapted where suitable. Also, the schemes are modified for more clarity, and reaction yields and stereoselectivities are added.    

  1. On a side note, and not entirely pertinent to this article, there are already work where chiral ligands have been ionically and covalently tethered to heterogeneous surfaces and authors have raised a good point highlighting this in the conclusion section that more work can be done in this area.

Answer: We appreciated kind consideration of our manuscript. Based on the concept of our work and fully revisions made to three reviewers’ comments, now we consider that this paper will be of interest to the many researchers reached by Catalysts.

Reviewer 3 Report

This manuscript consists of a standard review on organocatalytic methods for the synthesis of alpha-aminonitriles. The contents are correct and the presentation of data is sound and potentially useful to chemists working in this field.

                The authors should consider the following aspects prior publication:

1.       Page 3, lines 121-124: The definition of a chiral molecule is perhaps out of place in a review of this level. Readers already know what chirality is.

2.       Page 6, line 165: Instead of “study”, please write “method” or a similar term.

3.       Page 6, lines 173-185, Figure 5 and Scheme 2: The authors should emphasize that the metal (K+) at the active site of the organocatalyst is provided by the KCN reactant.

4.       Pages 8-10. Section “2.2. Synthesis of racemic alpha-aminonitriles” includes catalysts possessing W and Fe metallic centers and therefore is not adequate for this review.

Author Response

This manuscript consists of a standard review on organocatalytic methods for the synthesis of alpha-aminonitriles. The contents are correct and the presentation of data is sound and potentially useful to chemists working in this field.

Answer: Thanks for the positive comments. -aminonitriles are an essential class of chemicals with application in many fields, especially in pharmaceutical industries. We strongly believe this review will be a key summary for researchers in this area.

The authors should consider the following aspects prior publication:

  1. Page 3, lines 121-124: The definition of a chiral molecule is perhaps out of place in a review of this level. Readers already know what chirality is.

Answer: As suggested by the reviewer, the sentences were revised. We have removed the very general terms used throughout this document.

  1. Page 6, line 165: Instead of “study”, please write “method” or a similar term.

Answer: As suggested by the reviewer, the sentences were revised.

  1. Page 6, lines 173-185, Figure 5 and Scheme 2: The authors should emphasize that the metal (K+) at the active site of the organocatalyst is provided by the KCN reactant.

Answer: As suggested by the reviewer, the sentences were revised. A new sentence is added to the revised manuscript.

Page 6, lines 171-175

The metal (K+) present at the active organocatalyst site is provided by the KCN reagent. This method provided excellent yields and efficiency for the synthesis of asymmetric α-aminonitriles Strecker products from α-amido sulphones. Using this efficient catalytic protocol, enantiomerically pure non-natural a-amino acids were produced on a gram scale from α-amido sulphones.25

  1. Pages 8-10. Section “2.2. Synthesis of racemic alpha-aminonitriles” includes catalysts possessing W and Fe metallic centers and therefore is not adequate for this review.

Answer: In the case of Fe-based catalysts, the metal oxide center was used to design recyclable catalysts. However, the active sites are located on the organo sites (Tetrahedron Lett. 2013, 54 (48), 6403–6406). Such a strategy is often used to design the preparation of recyclable organocatalysts. Therefore, we consider this to be an important part of this review and keep it as is. Nevertheless, we removed the sulfated tungstate-based catalysts for the Strecker reaction.

The comments from three reviewers were most helpful and gave us a better perspective of our work. We feel that the present manuscript is greatly improved by the revisions made. Now we consider that this paper will be of interest to the many researchers reached by Catalysts.

Round 2

Reviewer 1 Report

The changes are fine and the manuscript can be accepted. However, the authors should ensure that there should not be any copyrights related issue i.e., copy-paste from other source without permission as they state in their responses.